# Alzheimer’s Disease, a Lipid Story: Involvement of Peroxisome Proliferator-Activated Receptor α

**DOI:** 10.3390/cells9051215

**Published:** 2020-05-14

**Authors:** Francisco Sáez-Orellana, Jean-Noël Octave, Nathalie Pierrot

**Affiliations:** 1Université Catholique de Louvain, Alzheimer Dementia, Avenue Mounier 53, SSS/IONS/CEMO-Bte B1.53.03, B-1200 Brussels, Belgium; francisco.saez@uclouvain.be (F.S.-O.); jean-noel.octave@uclouvain.be (J.-N.O.); 2Institute of Neuroscience, Alzheimer Dementia, Avenue Mounier 53, SSS/IONS/CEMO-Bte B1.53.03, B-1200 Brussels, Belgium

**Keywords:** Alzheimer’s, risk factors, PPARs, PPARα, lipids, fatty acids, modulators, cognition, sex, therapy

## Abstract

Alzheimer’s disease (AD) is the leading cause of dementia in the elderly. Mutations in genes encoding proteins involved in amyloid-β peptide (Aβ) production are responsible for inherited AD cases. The amyloid cascade hypothesis was proposed to explain the pathogeny. Despite the fact that Aβ is considered as the main culprit of the pathology, most clinical trials focusing on Aβ failed and suggested that earlier interventions are needed to influence the course of AD. Therefore, identifying risk factors that predispose to AD is crucial. Among them, the epsilon 4 allele of the *apolipoprotein E* gene that encodes the major brain lipid carrier and metabolic disorders such as obesity and type 2 diabetes were identified as AD risk factors, suggesting that abnormal lipid metabolism could influence the progression of the disease. Among lipids, fatty acids (FAs) play a fundamental role in proper brain function, including memory. Peroxisome proliferator-activated receptor α (PPARα) is a master metabolic regulator that regulates the catabolism of FA. Several studies report an essential role of PPARα in neuronal function governing synaptic plasticity and cognition. In this review, we explore the implication of lipid metabolism in AD, with a special focus on PPARα and its potential role in AD therapy.

## 1. Alzheimer Disease: A Dementing Illness

With a prevalence doubling every 5 years beyond 65, Alzheimer’s disease (AD) is a devastating neurodegenerative disorder, which is the most common cause of dementia in the elderly. In 2019, Alzheimer’s Disease International estimated that there are over 50 million people living with dementia globally, a figure set to increase to 152 million by 2050 [1].

Memory loss is one of the main clinical features of AD onset, and numerous neuropsychological tests allow for the assessment of the cognitive functions of Alzheimer’s patients [2,3]. Along with a decline in cognitive performance, AD is characterized by the coexistence in the brain of two main neuropathological lesions: intraneuronal neurofibrillary tangles composed of hyperphosphorylated microtubule-associated protein tau and extracellular senile plaques containing the amyloid-β (Aβ) peptide generated from the sequential proteolytic processing of its precursor, the amyloid precursor protein (APP). Although the definitive diagnosis of the disease was previously achieved by the postmortem neuropathological brain examination, the detection of specific AD biomarkers in the cerebrospinal fluid, including Aβ and tau, constitutes an early examination and a reliable diagnosis [4]. Moreover, recent non-invasive imaging techniques using Aβ- and tau-PET tracers have led to the preclinical diagnosis of AD, allowing its evolution during the patient’s lifetime to be tracked [5,6,7]. As positron emission tomography (PET) imaging studies have shown that Aβ accumulation occurs long before the onset of clinical AD and given that mutations in the *APP* gene can act as fully penetrant in rare inherited early-onset AD cases (EOAD, about 1% of the cases, (Figure 1)), the amyloid cascade hypothesis was proposed to explain the pathogeny. According to this hypothesis, a gradual accumulation and aggregation of Aβ initiate a neurodegenerative cascade resulting in neurofibrillary tangles formation, cell loss, vascular damage and dementia [8,9]. Although this hypothesis was strengthened by the discovery that mutations in presenilins (Figure 1), the catalytic subunits of the γ-secretase complex, lead to an increase in Aβ production [10], several studies have challenged the amyloid hypothesis over the past ten years [11,12]. Mounting evidence reports that mutations in the *Presenilin 1* (*PSEN1*) gene associated with EOAD have heterogenic manifestations. It was indeed recently shown that the most common mutations found in *PSEN1* decrease the activity of the γ-secretase [13,14] or lead to a loss of its function [15], indicating that in some cases, *PSEN1* mutations either hyper-activate or reduce the activity of the γ-secretase complex. Moreover, γ-secretase activity assessed in brain samples from EOAD and non-demented controls was similar, while it displayed some dysfunctions in a few brain samples from late-onset AD cases (LOAD), which represent the vast majority of AD cases [14]. This suggests that γ-secretase may also play a role in some LOAD cases, in which brain Aβ production levels are similar to those observed in unaffected controls [16]. 

Moreover, several studies have shown that humans with Down syndrome, who harbor three copies of the *APP* gene that leads to the overexpression of APP protein, have an age-dependent increased risk for developing AD and develop clinical features and neuropathological changes similar to those observed in AD (for review, see [17]). This aforementioned study suggests that AD could be a combination of different pathologies with diverse etiologies [18] leading to dementia. This is supported by recent findings showing that some pathologies have similar clinical markers and manifestations to those observed in AD, as reported in limbic-predominant, age-related TDP-43 encephalopathy (LATE), in which senile plaques and neurofibrillary tangles that define AD have been brought out in the brain [19]. 

Since Aβ that builds up in plaques also deposits during normal brain aging, amyloid deposition occurring in the hippocampus and cerebral cortex of AD patients potentially explains deficits in memory and cognitive function observed. Despite the various isoforms of Aβ produced, Aβ toxicity rate is dependent on its state of assembly. Among the three assemblies state of Aβ (monomers, soluble oligomers and insoluble fibrils) (for review, see [20]), soluble Aβ oligomers are organized into different structures ranging from dimers, trimers, tetramers, pentamers, decamers and dodecamers, among others [21,22,23]. Toxic soluble oligomers have been identified in AD brains [24,25,26]. However, in some cases, higher aggregates, such as fibrils, showed protective effects in AD models [27], suggesting that there is an inverse correlation between the size of Aβ assembly and its toxicity. Since Aβ dimers form a more stable structure, these dimeric units are described to be the building blocks for toxic aggregates [28]. This supports the idea that disassembling plaques or fibrillar structures could be detrimental if not accompanied by strategies to remove oligomeric aggregates of Aβ. However, it remains difficult to influence the course of AD by removing amyloid deposits. Indeed, some approaches were developed to inhibit secretases activities involved in the release of Aβ or to remove amyloid deposits from the brain using active or passive immunotherapy [29]. While first attempts completely failed in Phase III clinical trials due to the widespread function of secretases and the development of encephalitis in some patients [30], recent attempts have shown that although senile plaques can be effectively removed from the AD brain, cognitive performance is not improved in these patients [31]. Despite the fact that recent results with an immunotherapy clinical trial using aducanumab targeting aggregated forms of Aβ have been encouraging and could prove efficacious [32], this clearly suggests that when structural modifications are found in the brain due to accumulation of abnormal proteins, the proposed treatments arrive too late and are inefficient [33]. Consequently, we must focus on modifications in physiological functions, which could occur long before abnormal protein deposition. Among them, abnormal lipid metabolism could be an important early event in the pathogenesis of AD. 

## 2. Linking Lipids to Alzheimer’s Disease

While in EOAD, Aβ accumulation in the brain is caused by gene mutations, the vast majority of AD cases are late-onset AD cases (LOAD, 99% of cases), in which the source of Aβ accumulation in the brain is still unknown. Nevertheless, it is well established that the epsilon 4 allele of the *Apolipoprotein E* (*APOE*) gene, encoding the main lipid carrier in the brain, is a genetic risk factor for AD. People who are homozygous for this allele have ten times greater risk to develop AD [34]. Therefore, a relationship between AD and lipid metabolism has been established. Furthermore, large-scale genome-wide association studies on AD first confirmed that APOE4 is a major risk factor and provided evidence that at least 20 genetic susceptibility loci in addition to *APOE* genotype are associated with AD [35,36] (Figure 2). Among them are genes encoding Clusterin and ABCA7, two proteins involved in lipid metabolism [37,38,39,40] (Figure 1 and Figure 2). Therefore, the identification of these susceptibility loci supports the hypothesis that perturbation of lipid metabolism favors the progression of AD [41]. This hypothesis is sustained by recent reports showing that genetic polymorphisms in *SREBF* genes encoding sterol regulatory element-binding proteins (SREBPs), transcription factors activating lipid metabolism-related genes involved in cholesterol and fatty acids biosynthesis [42,43], were associated with an increased risk of schizophrenia and LOAD [44,45,46]. Disturbances in the signaling and expression of SREBPs were indeed reported in LOAD cases and, in a rare case of EOAD, harboring a microduplication in the locus of *APP* gene [47,48].

Additional support comes from metabolomic studies that have shown changes in the lipid content of plasma, cerebrospinal fluid and in brain tissue from AD [49,50,51,52]. Moreover, perturbations in brain fatty acids profiles observed in brain regions vulnerable to AD pathology [49,53] could influence AD pathogenesis by promoting Aβ accumulation and tau pathology [54,55,56].

## 3. Lipids in Alzheimer Disease: Involvement of Fatty Acids in Cognitive Function

The human brain contains the second-highest concentration of lipids (50–60% of its dry weight) after adipose tissue [57]. Due to their structural diversity and their involvement in a wide range of biological processes, lipids play a fundamental role in maintaining brain physiological functions. Phospholipids, sterols, sphingolipids, fatty acids and triacylglycerols are the five main brain lipid classes, which are involved in neuronal differentiation, synaptogenesis, and brain development (for reviews, see [58,59]). At the subcellular level, lipids are basic structural components of cell membranes and are enriched in the myelin sheath surrounding nerve cell axons, which regulates the ability of a neuron to trigger action potentials encoding information [60]. Moreover, lipids and their derivatives modulate membrane fluidity and permeability, which regulate trafficking, localization and function of ion pumps, channels, receptors and transporters at the plasma membrane [61,62]. In particular, any change in lipid homeostasis affects the lipid composition of membrane lipid rafts, which are cholesterol and sphingolipid-enriched microdomains [63] where most of the synaptic-related proteins involved in synaptic transmission and plasticity are embedded [64]. Since lipid rafts from AD brains displayed important changes in their composition [65,66], disturbance in the function of lipid-rafts-associated synaptic proteins could contribute to the development of neuropathological events that favor amyloidogenesis and proteins aggregation [67,68].

One of the main pathological characteristics involved in the pathogenesis of AD implies the APP protein, which is hydrolyzed by β- and γ-secretases, leading to the deposition of Aβ in the brain. APP is a single-pass transmembrane protein with a large extracellular region that contains several domains involved in APP dimerization, protein–protein or metal interactions (e.g., heparin- and copper-binding domains). The APP trans-membrane-helix domain in which the Aβ sequence is inserted and the APP intracellular domain contain cholesterol-binding [69] and YENPTY motifs that regulate the subcellular location, trafficking, and proteolytic processing of APP, respectively [70] (for more details, see [71,72]). These domains and motifs engage APP and its cleavage products in a plethora of physiological functions ranging from synaptic transmission, plasticity, development, neuroprotection, trophic function, cell adhesion, apoptosis, calcium and lipid homeostasis, among others [73,74,75].

The cholesterol-binding motif found in APP plays an essential role in the interaction of APP with proteins involved in cholesterol metabolism (e.g., SREBP1) [48] and in its location in lipid rafts present in synaptic vesicles and mitochondria-associated endoplasmic reticulum (ER) membranes (MAMs) [76] (Figure 1). MAMs are enriched in cholesterol and sphingomyelin and are points of physical contact between the outer mitochondrial membrane and ER. While they play an essential role in the metabolism of glucose, phospholipids, cholesterol and calcium homeostasis [77,78], MAMs regulate APP processing. Indeed, presenilins and γ-secretase activity, previously localized at the ER, are enriched in MAMs, in which β-cleaved fragment generating Aβ accumulates [79,80]. Interestingly, the activity/function and expression of MAM-associated proteins increase in human and mouse AD brains long before Aβ deposition [81], suggesting a potential role of MAMs in the pathophysiology of AD [81,82]. From these data, the concept of the MAM hypothesis in AD emerged (reviewed in more details in [83,84]).

While rafts are described as noncaveolar lipid microdomains, caveolae are cholesterol-enriched membrane invaginations found in the Golgi network, exocytotic vesicles, ER and plasma membrane in which surface protein markers caveolin are embedded [85,86]. Caveolae are involved in cellular cholesterol transport and are docking sites for signaling proteins and receptors and are therefore considered as hotspots for cell–cell communication [87]. Although caveolae-dependent cell signaling is not yet fully understood, several studies have reported the involvement of caveolin proteins in the pathogenesis of AD [88,89]. Caveolin expression levels are upregulated in the hippocampus and the frontal cortex of AD brain compared to control, suggesting a link between the expression of caveolin and dysregulation of cholesterol homeostasis observed in AD [89,90,91]. Moreover, increased expression in caveolin promotes oxidative stress and APP processing into Aβ [89,92] that could favor the progression of AD. Although cholesterol and sphingolipid-enriched membrane microdomains could take part in AD physiopathology, fatty acids seem to also contribute to its occurrence.

Fatty acids (FAs) are the major essential monomeric constituents of all lipids [93] and therefore are key components of cellular membranes [94]. They can be unesterified (free) or esterified to plasma membrane phospholipids and are classified based on the length of their carbon chain. FAs are either saturated, monounsaturated, or polyunsaturated (PUFAs) (for review, see [95]). While the brain can produce saturated and monosaturated FAs by de novo lipogenesis, essential PUFAs cannot be synthesized in sufficient quantities [96] and therefore are provided by the diet [94]. Within the brain, palmitic acid and stearic acid are the main saturated FAs, and oleic acid is the main monounsaturated one. Linoleic and α-linolenic essential FAs are transformed into arachidonic and docosahexaenoic acids, the major brain ω-6 and ω-3 PUFAs, respectively [94,95].

PUFAs play a critical role in neurogenesis, synaptic function, inflammation, glucose homeostasis, mood and cognition [94]. As they play a critical role in brain development and functioning [97], high concentrations of dietary saturated long-chain FAs and a decrease in dietary consumption of ω-3 PUFAs have been associated with neurological dysfunction and neuropsychiatric disorders, including neurodegenerative diseases such as AD [98,99,100]. In addition, diets in the western population are rich in saturated FAs and low in PUFAs [101], which are not only associated with the development of obesity but also to cognitive dysfunction.

Levels of docosahexaenoic acid (DHA), the major brain ω-3 PUFAs, have been reported to be decreased in plasma and post-mortem brains from AD patients [49,50,51]. Although DHA dietary supplements did not improve memory, cognition or mood [50,102,103], higher dietary intake of DHA is associated with decreased risk of neurological disorders [104] and dementia in elderly individuals [105]. Interestingly, ω-3 PUFAs supplementation in individuals with mild cognitive impairment and in AD patients without the *APOE4* allele has shown benefits [102,106]. While low brain DHA levels were shown to impair behavior in AD mouse models [107,108], the dietary supplementation of ω-3 PUFAs in rodents facilitated hippocampal synaptic plasticity and improved cognitive deficits of aged mice and in several animal models of AD [50,102,109,110,111,112]. Moreover, in non-pathological conditions, maternal intake of ω-3 PUFAs increases hippocampal plasticity and cognition in healthy pups rodents [113].

While cellular and molecular mechanisms underlying such effects are poorly understood, more and more studies put forward the involvement of nuclear receptors.

## 4. RXRs, LXRs and PPARs Nuclear Receptors in AD

### 4.1. Nuclear Receptors

The nuclear receptors superfamily of ligand-dependent transcription factors regulates energy balance, inflammation, and lipid and glucose metabolism [114]. They control target genes expression through their binding with sequence-specific elements located in gene promoter regions [114]. Structurally, they contain an amino-terminal activation domain needed for the recruitment of coactivators, a carboxyl-terminal ligand- and a DNA-binding domain. Among these receptors, retinoid X receptors (RXRs), liver X receptors (LXRs) and peroxisome proliferator-activated receptors (PPARs) act as master regulators of lipid metabolism by *trans*-activating genes encoding enzymes involved in lipid and fat metabolism. Therefore, they are abundantly expressed in metabolically active tissues, including the brains of rodents and humans [115]. Due to their anti-inflammatory and potential neuroprotective effects, RXRs, LXRs and PPARs activation with specific agonists emerged as promising approaches for treating brain pathologies in several mouse models of Parkinson’s, Huntington and Alzheimer’s diseases, multiple and amyotrophic lateral sclerosis, stroke and even in a mouse model with physiological brain aging-dependent cognitive decline (reviewed in [116,117]).

### 4.2. RXRs

Among the three RXR isotypes identified (RXRα, β and γ), RXRα is mainly expressed in the liver, lungs, muscles, kidneys, epidermis and intestine. While RXRβ is expressed ubiquitously, RXRγ is enriched in the brain, heart and muscles. RXRs can be activated by 9-*cis* retinoic acid, linoleic, linolenic and DHA acids, natural RXR ligands [118,119]. As a strong agonist of the RXRs, the retinoid bexarotene synthetic agonist [120], which the U.S. FDA approved for the treatment of cutaneous T-cell lymphoma [121], was described to improve cognitive deficits in AD mouse models [122,123,124,125,126,127] mainly by inducing the transcription and lipidation of APOE and reducing microglial expression of pro-inflammatory genes among others [128,129]. Although we previously reported that bexarotene improved cognition in a patient with mild AD [130], its efficacy in clinical trials for treating AD pathology has been disappointing [131,132].

### 4.3. LXRs

As an obligate binding partner of LXRs, RXRs form permissive heterodimers with two LXRs isoforms, LXRα and β [114]. LXRα is abundantly expressed in the liver, intestine, kidney, spleen and adipose tissue, whereas LXRβ is ubiquitously expressed at a lower level but more widely in the brain and mainly in the hippocampus. LXRs are activated by oxysterols, most prominently hydroxylated forms of cholesterol [133,134]. They play therefore a critical role in the control of whole-body cholesterol homeostasis and exert potent anti-inflammatory actions [135]. Once activated, they control the transcription of target genes involved in lipid transport and biosynthesis, such as APOE and SREBP, respectively [136,137]. The expression of the SREBP1 isoform is mediated by LXRs to ensure FAs synthesis needed for the esterification of free cholesterol for protecting cells from a detrimental cholesterol overload. Moreover, unesterified PUFAs exert feedback inhibition on the expression of SREBP1 by antagonizing the oxysterol LXR receptor [138,139].

### 4.4. PPARs

PPARs were first described for their ability to induce peroxisomal proliferation in the liver in response to xenobiotics [140]. Afterward, they were considered as master metabolic regulators involved in energy homeostasis [141]. They act principally as lipid sensors and regulate whole-body metabolism in response to dietary lipid intake and control their subsequent metabolism and storage by inducing or repressing the expression of genes involved in the metabolism of lipid and glucose [142]. The three PPARs isoforms identified (PPARα, β/δ and γ) have partially overlapping functions and tissue distribution in mammals. PPARα is highly expressed in the liver, heart and kidney but has low levels in the brain [143,144] and more particularly in the hippocampus of rodents and primates [144,145,146,147,148,149]. PPARα plays an important role in the regulation of FAs catabolism [150] by controlling the expression of genes encoding acyl-CoA oxidase, carnitine palmitoyl transferase and acetyl-CoA carboxylase, enzymes that tightly regulate FAs peroxisomal and mitochondrial β-oxidation, respectively (for reviews, see [151,152]) (Figure 1). Consistent with the first central role of PPARα in FAs catabolism [150], PPARα null mice exhibit greater lipid accumulation [153].

While PPARγ isoform is mainly expressed in white and brown adipose tissue, the large intestine and spleen, in which it regulates adipogenesis, energy balance, lipid biosynthesis and inflammation [154], PPARβ/δ is expressed ubiquitously in all tissues and is the most abundant isoform found in liver, kidney, adipose tissue and skeletal muscle, where it plays mainly a role in FAs oxidation [155].

Although PPARs expression is ubiquitous in the human and mouse brain, PPARα and γ are expressed in both neurons and astrocytes, while PPARβ/δ isoform is exclusively neuronal [115].

Although PPARs were first classified as orphan receptors, many natural and synthetic agonists of PPARs are used in the treatment of glucose and lipid disorders. Several endogenous ligands from dietary lipids and their metabolites were identified, among them the essential FA DHA and eicosanoids [156,157]. Recently, hexadecanamide, 9-octadecanamide and 3-hydroxy-(2,2)-dimethyl butyrate have been identified as endogenous PPARα ligands in mouse brain hippocampus [158]. Moreover, several synthetic ligands are widely used in clinical practices, among them fibrates and thiazolidinediones, PPARα and γ agonists, used in the treatment of hypertriglyceridemia and diabetes mellitus, respectively (for review, see [159]). In addition, PPARs ligands [160] decrease Aβ burden, tau phosphorylation and inflammation and improve behavior in AD mouse models [116,161].

Due to their overlapping expression in all brain cell types from mouse and human and given that they share similarities in their ligand-binding domains [162,163], a tight interconnection between PPARs isoforms was described a couple of years ago [164,165]. The mutual interactions observed between PPARα, β/δ and γ lead to the concept of a “PPAR triad” in the brain (reviewed in [166]). This concept emerged from data reporting that the activation of a PPAR isoform affects the expression of other PPARs due to the low isotype specificity of endogenous PPAR ligands [166,167]. Indeed, the simultaneous activation of different PPARs isoforms was first shown in C6 glioma and lipopolysaccharide (LPS)-stimulated astrocytes, in which the activation of PPARβ/δ increases the expression of PPARγ and to some extent that of PPARα in a positive feedback loop [164,165]. Moreover, such crosstalk between PPARs was also reported in primary cortical neurons and in ischemic rat brain, where PPARγ activation stimulates the interleukin-1 receptor antagonist production through the activation of the PPARβ/δ [168]. Conversely, PPARα agonist reduces the expression of PPARβ/δ in LPS-stimulated primary cultures of astrocytes in a negative feedback loop, leading to the downregulation of the cyclooxygenase (COX)-2 enzyme involved in the synthesis of the endogenous PPAR agonist prostaglandin [164,166]. While the activation of PPARα represses the expression of COX-2, PPARβ/δ activation upregulates the expression of both COX-2 and cytosolic phospholipase A2, producing PUFAs [164,166] (Figure 2). Therefore, the cross-talk between PPAR isoforms highlights a “PPAR network”, in which the activation of each PPAR participates in the fine-tuning of genes expression. These interconnections between PPARs should be considered to design appropriate therapeutic strategies for neurodegenerative disorders, including AD.

## 5. PPARs in Alzheimer’s Disease Therapy: The Promising Role of PPARα

### 5.1. PPARγ and PPARβ/δ in AD

Considering that AD and metabolic diseases such as obesity and type 2 diabetes have overlapping metabolic dysfunctions (e.g., dyslipidemia, glucose metabolism impairment and insulin resistance) and given that PPARs metabolic regulators are expressed in the brain, it is not surprising that changes in PPARs signaling might lead to dementia [169,170,171,172].

Since PPARγ and PPARβ/δ regulate both lipid and carbohydrate metabolism and insulin sensitivity, these receptors represent an attractive therapeutic target for AD. The reduction in glucose metabolic rates observed in the AD brain occurs decades before onset of clinical symptoms and supports the idea that metabolic deficits are upstream events, which may influence the course of AD [173]. As a defining feature of AD, brain glucose hypometabolism leads to a decrease in the O-GlcNAcylation (O-GlcNAc) of proteins, including both tau and APP. While an increase in brain O-GlcNAc protects against tau and Aβ peptide toxicity, a decrease in O-GlcNAc promotes neurodegeneration [174].

Moreover, brain insulin resistance promotes AD pathophysiology by disrupting energy homeostasis and insulin signaling pathways [175,176]. Impairment in insulin signaling favors Aβ-mediated oxidative stress, Aβ secretion, brain amyloid deposition and tau pathology (reviewed in [177,178,179]). Therefore, targeting PPARβ/δ and PPARγ with specific drugs represents an effective strategy to preserve carbohydrate metabolism, insulin-sensitizing pathways and cognitive performance. By far, PPARγ was first considered as a promising target for the treatment of AD. While the thiazolidinedione class of PPARγ agonists has shown improvement in cognitive behavior in murine models of AD [161,180,181], human clinical trials using PPARγ agonists are less encouraging [182,183]. Although the chronic treatment of diabetic patients with the PPARγ agonist pioglitazone reduces dementia risk by 47% [184], Takeda and U.S. partner Zinfandel Pharmaceuticals decided to give up and stop testing a 20-year-old diabetes medicine that fails once more in AD therapy, a lack of success attributed to the low penetrance of glitazones in the brain.

In contrast to PPARγ, PPARβ/δ is highly expressed throughout the brain and therefore represents a new therapeutic target of interest in AD [185]. Indeed, treatments using PPARβ/δ agonists have been reported to decrease brain neuroinflammation, neurodegeneration, amyloid burden and improve cognitive function in several AD mouse and rat models [186,187,188,189]. Recently, a Phase IIa clinical study of the dual PPARδ and γ agonist T3D-959 reports plasma metabolome profile changes on lipid, glucose and insulin-related metabolism and improvements of cognitive function (presumably associated with *APOE* genotype) in a small cohort of patients with mild to moderate AD [190].

### 5.2. PPARα in AD

Although the function of PPARα in the brain remained elusive for a long time, more and more studies indicate that PPARα is involved in physiological and pathological brain functions (e.g., in the sleep-wake cycle [191,192], depression [193,194,195,196], epilepsy [197,198,199], stroke [200,201,202,203] and schizophrenia [204]). PPARα modulators (e.g., oleoylethanolamide, a natural PPARα ligand; Wy14643 and fibrates, two synthetic PPARα agonists) regulate dopamine and hippocampal brain-derived neurotrophic factor (BDNF) signaling pathways to rescue depression-related behaviors [193,195,196] and nicotinic acetylcholine receptors and endocannabinoid signaling to alleviate epilepsy and schizophrenia-like effects in mice [197,204].

#### 5.2.1. PPARα Function in Brain Energy Supply

In addition to their anti-inflammatory and potential neuroprotective effects [117,172,205,206,207], PPARs, in particular PPARα, are master metabolic regulators of energy homeostasis [141]. Several studies report that PPARα plays an essential role in maintaining brain energy supply. Ketone bodies, which are derived from FAs oxidation, are mainly produced in the liver during prolonged fasting or starvation and represent a significant alternative source of fuel to compensate for a lack of glucose in the brain [208,209,210]. The ketogenic diet has been therefore used in the treatment of several neurological diseases, including Parkinson’s and Alzheimer’s diseases, traumatic brain injury and epilepsy (reviewed in [211]). More and more evidence indicates that the ketogenic diet shows benefits in both in vitro and in vivo AD models. Treatment with the ketone body d-β-hydroxybutyrate protects hippocampal neurons from Aβ toxicity [212] and ketogenic diet decreases brain amyloid pathology in a mouse model of AD [213]. Moreover, the oral administration of the ketogenic compound AC-1202 reduces oxidative stress and inflammation and improves cognitive function in mild to moderate AD patients [214].

#### 5.2.2. PPARα and Cognitive Function

More recently, an essential role of PPARα in cognition has emerged. By using a passive-avoidance task, Mazzola et al. reported that memory acquisition is enhanced in rats treated with the PPARα agonist Wy14643 [215]. Moreover, treatment of mice with the Wy14643 attenuates cognitive impairments induced by scopolamine, a muscarinic acetylcholine receptor antagonist [216]. Consistent with the potential role of PPARα in cognition and memory, PPARα-deficient mice have spatial learning and long-term memory deficits [149], indicating that PPARα is required for normal cognitive function [217]. Roy et al. have shown that PPARα, and not PPARγ and PPARβ/δ isoforms, regulates the expression of a set of synaptic-related proteins involved in excitatory neurotransmission, including BDNF, GluN2A and GluN2B subunits containing N-methyl-D-aspartate receptors (NMDARs) and GluA1 subunit containing alpha-amino-3-hydroxy-5-methyl-4-isoxazolepropionic acid receptors (AMPARs) [149]. In agreement with this, we have recently reported that the absence of PPARα severely impairs hippocampal long-term potentiation (LTP), which is defined as an activity-dependent enhancement of synaptic strength involved in memory processing [218], and GluA1 expression in male mice [125].

Moreover, Roy et al. identified a PPAR-responsive element in the promoter of genes encoding the cAMP response element-binding (CREB) protein and therefore identified it as a PPARα target [149]. Interestingly, recent data indicate that RXR activation induces neuronal CREB signaling and increases dendritic complexity and branching of neurons promoting their differentiation and development [219,220]. In addition, activation of RXRs upregulates the expression of synaptic markers and improves cognition in a mouse model of AD [124]. Altogether, these data indicate that effects mediated by RXR activation on the expression of synaptic-related proteins and cognition could be PPARα-mediated.

It is also interesting to note that Chikahisa et al. recently reported that PPARα-null mice exhibit an enhancement of fear learning [221]. This enhancement results from an increase in levels of dopamine and its metabolites in the amygdala [221], suggesting that PPARα is likewise involved in the regulation of emotional memory via the dopamine pathway in the amygdala.

#### 5.2.3. Potential Link between PPARα and AD

The relevance of a potential beneficial effect of PPARα for dementia is supported by some studies showing that polymorphisms in *PPARA* gene encoding PPARα were associated with an increased risk of AD. In 2003, Brune et al. were the first to report an association of the *PPARA* L162V polymorphism with the AD risk [222]. They indicate that this risk is even higher in carriers harboring a polymorphism in *INS* gene encoding insulin [222]. The interaction of *INS* and *PPARA* genes in AD was thereafter investigated by Kölsch and colleagues [223]. In their study, they report an interaction on AD risk between *PPARA* L162V and *INS* in Northern Europeans, in whom Aβ42 and pro-inflammatory cytokines levels were increased in the cerebrospinal fluid (CSF) [223,224]. However, Sjölander et al. later reported a lack of replication of these studies [225]. They did not find significant differences in genotype or allele distributions between AD patients and controls and found no influence of *PPARA* variants on CSF markers [225]. Although these conflicting results question the promising role of PPARα in AD therapy, previous results indicate that expression levels of PPARα and β/δ are significantly reduced, whereas PPARγ expression is selectively increased in AD brains [226], indicating that PPARs function is impaired in AD and therefore may contribute to the progression of the disease.

#### 5.2.4. PPARα Ligands and AD

More and more studies report the beneficial effects of several PPARα synthetic agonists on cognitive behavior in several AD mouse models. Among them, fibrates (e.g., fenofibrate, bezafibrate, ciprofibrate and gemfibrozil) are a class of lipid-lowering drugs used in the treatment of metabolic syndromes, including hypertriglyceridemia, obesity and type 2 diabetes, which prevents the progression of atherosclerotic lesions, cardiovascular events and non-alcoholic fatty liver disease (reviewed in [227,228]). Among fibrates, fenofibrate has been widely used, but its relatively low efficiency as PPARα agonist [229,230] leads to the development of pemafibrate, a more potent and selective agonist for PPARα [231,232,233]. Recent results from two Japanese Phase III clinical studies indicate that pemafibrate improves lipid profiles in patients with type 2 diabetes and hypertriglyceridemia [234] and was useful for dyslipidemia, with a much higher efficacy than fenofibrate [235].

The salutary effects of fibrates on memory have been reported in several preclinical AD models. It was recently demonstrated that administration of the PPARα activator gemfibrozil decreases amyloid plaque burden, microgliosis and astrogliosis in the hippocampus and cortex of 5XFAD mice [236], a well-characterized transgenic mouse model of AD, in which age-dependent synaptic and cognitive deficits occur [237]. Although a decrease in the expression of PPARα was observed in the brain of 5XFAD mice [125,236], oral administration of gemfibrozil or pemafibrate improves spatial learning, memory and hippocampal LTP, respectively, in these mice [125,236].

More recently, Luo et al. reported that amyloid pathology, memory deficits and anxiety were reversed in the APP-PSEN1ΔE9 mouse model of AD treated with either gemfibrozil or Wy14643 [238]. The effects observed were mediated by a PPARα-dependent enhancement of autophagosome biogenesis [238].

In addition, the activation of PPARα with non-conventional ligands such as statins or aspirin, cholesterol-lowering and nonsteroidal anti-inflammatory drugs, improves [239] hippocampal plasticity and memory in 5XFAD but not in 5XFAD/*Ppara*-null mice by mediating the transcriptional activation of BDNF and CREB, respectively [239,240]. Moreover, oral administration of low-dose aspirin decreased amyloid plaque pathology in 5xFAD mice by stimulating PPARα-mediated lysosomal biogenesis [241].

Overall, these results support the potential for using PPARα ligands as a promising strategy for the treatment of AD.

Among PPARα ligands, gemfibrozil has recently been assessed as a possible treatment for AD. Although gemfibrozil has first been tested in a Phase I clinical trial (NCT00966966 [242]) in healthy volunteers to evaluate its safety and absorption (unpublished data), a second early Phase I trial (NCT02045056 [243]) is testing its efficiency to prevent AD by evaluating its ability to increase microRNA107 (mir-107) levels in participants with either no or mild cognitive impairment. It was previously shown that expression of mir-107, a small noncoding RNA involved in the regulation of gene expression [244], is reduced in AD and may accelerate disease progression through the regulation of β-Site amyloid precursor protein-cleaving enzyme 1 (BACE1) [245,246], an endopeptidase that cleaves APP to generate Aβ [247,248]. Moreover, gemfibrozil-mediated activation of PPARα has been reported to promote the non-amyloidogenic APP processing [249]. In 2015, Corbett et al. identified PPAR-responsive elements in the promoter of the gene encoding the α-secretase “a disintegrin and metalloproteinase domain-containing protein 10” (ADAM10), a new PPARα target [249] (Figure 2). APP cleavage by ADAM10 precludes Aβ generation and results in the release of a soluble APP α (sAPPα) fragment which exerts neurotrophic and neuroprotective properties involved in the maintenance of dendritic integrity in the hippocampus [250]. Treatment of wild-type mouse hippocampal neurons with gemfibrozil increases sAPPα and decreases Aβ production [249]. Moreover, the production of brain endogenous Aβ is increased in PPARα-deficient mice and exacerbated in 5XFAD/*Ppara*-null when compared to wild-type and 5xFAD respective littermates [249].

Although PPARα is indubitably involved in the non-amyloidogenic processing of APP, PPARγ was previously demonstrated to also regulate Aβ production by controlling the expression of *Bace1* gene. PPARγ activation with specific agonists (e.g., thiazolidinediones and non-steroidal anti-inflammatory drugs including ibuprofen) decreases the expression of BACE1 [251], whereas a lack of PPARγ has an opposite effect in cultured mouse embryonic fibroblasts [252], suggesting that PPARγ is a repressor of *Bace1*.

#### 5.2.5. PPARα and Sex 

While sexual dimorphisms of PPARγ agonist rosiglitazone were previously reported on insulin sensitization and glucose in mice [253], most in vivo studies have analyzed the potential effects of PPARα modulators on cognition mainly in male and not in female rodents. We have previously reported a sex-regulated gene dosage effect of PPARα on synaptic plasticity [125]. PPARα activation improves synaptic plasticity only in male but not in female 5XFAD mice [125]. These observations were concomitant with a higher expression of PPARα in brains of males as compared to females [125]. Such differences in PPARα expression between male and female rodents were previously reported in liver [254], lymphocytes [255] and ischemic brain [202]. Moreover, a sexual dimorphism was also observed in hippocampus-dependent behaviors. Numerous studies have previously reported that the magnitude and maintenance of LTP were larger in males than in females, not only at CA3-CA1 synapses but also in the dentate gyrus-CA3 and temproammonic-CA1 synapses of the hippocampus [256,257,258,259,260,261].

The most obvious difference between males and females is sexual hormones. Hormones are known to influence the expression of PPARα in a sex-specific manner since gonadectomy of male rats decreases PPARα expression [254]. Estrogens, such as estradiol, are known to improve synaptic plasticity [262,263], and behavior is affected in ovariectomized female rats [264,265,266]. In humans, cognitive impairments in older women have long been attributed to the decrease in circulating estradiol levels after menopause. Exogenous restitution of this hormone during the perimenopausal period ameliorates such impairments [267,268]. In addition, estrogen replacement therapy in women in a specific time window is associated with reduced incidence of AD (reviewed in [269]). Although no differences in PPARα expression were reported between men and women in skeletal muscles [270], several studies indicate that human circulating mononuclear and T cells exhibit sex differences driven by the expression of PPARα and PPARγ [271,272]. In their study, Zhang et al. showed that the treatment of T cells with androgens increases PPARα and decreases PPARγ expression [272].

It is known that women are at a higher risk for AD (two-thirds of those with AD are women). This results partly from differences between men and women in life-expectation and biology (e.g., sex-specific differences in gene expression, hormone levels, brain structure and function and in inflammatory response) (reviewed in [273,274]). Such differences are not exclusively related to AD but are also observed in cardiovascular diseases, metabolic syndromes and diabetes, where postmenopausal women or women with endocrine disorders (e.g., in Polycystic Ovary Syndrome or Primary Ovarian Insufficiency, in which levels of androgens or estrogens are increased or decreased, respectively), are at higher risk to develop these pathologies when compared to non-affected women (reviewed in [275]). This suggests a potential role for estrogens in metabolic function and in particular in brain metabolism. Indeed, impairment in estrogenic regulation of brain glucose metabolism was previously reported during perimenopause [276,277], and brain hypometabolism reported in women in menopausal transition is associated with cognitive dysfunctions [278,279]. This could result from a decrease in the activation of estrogen receptors (members of the superfamily class of nuclear receptors) in brain areas involved in learning and memory processes, including the prefrontal cortex and hippocampus [280]. Moreover, impairments in the estrogenic regulation of mitochondrial bioenergetics [281] could lead to subsequent oxidative stress, promoting Aß accumulation and neuronal dysfunction [282]. Furthermore, sex differences in PPARs expression and function reported in rodent and human brain could result from changes in sex hormones levels and cross-talks between estrogen receptors and PPARs network (reviewed in [283]), suggesting a role for these receptors in sexual dimorphisms observed in metabolism, inflammatory response and brain function. Moreover, sex differences of the plasma and brain lipidome have been mentioned in humans [284,285,286,287], supporting a potential role of PPARα in this aspect.

## 6. Conclusions

AD is a multifactorial neurodegenerative disorder in which cognitive deficits occurred. AD is influenced by genotype and environmental factors. Among risk factors identified, genomic loci encoding proteins involved in lipid metabolism and altered lipidome of AD brain suggest that brain lipid metabolism is impaired in AD. Moreover, obesity and type 2 diabetes metabolic disorders, identified as AD risk factors, support the essential role of lipid homeostasis in the etiology of AD. Interestingly, PPARs are nuclear transcription factors that govern pathways implicated in the etiology of AD, including lipid metabolism and inflammatory response (Figure 2). Among them, PPARα involved in the catabolism of FAs plays a crucial role in cognitive brain function. While disease-modifying treatments for AD are seeking to interfere with the pathogenic steps responsible for clinical symptoms, PPARs modulators are a promising target in AD therapy. Among PPARs, PPARα has a particular interest since it is the only PPAR isoform described to have neuronal functions involved in memory processes.

As the expression of PPARs is modified in the AD brain, the characterization of new synthetic molecules able to activate several PPARs isoforms could be needed for an efficient treatment for AD. Alternative strategies could be therefore to design novel pan-agonists that can simultaneously activate PPARα, PPARβ and PPARγ. Their efficiencies were previously demonstrated in preclinical mouse model of AD and therefore deserve further investigation.

Since a sexual dimorphism of PPARα agonist was observed in mechanisms underlying memory processes in vivo, sex differences should be considered in therapy aiming to use PPARα modulators. In humans, the influence of sex on the incidence, manifestation and treatment of numerous neurological and psychiatric diseases is well documented. Therefore, it is of crucial importance to decipher sex differences in AD, in which complex cognitive and neuropsychiatric symptoms occur, in order to define novel PPARα sex-specific therapeutic strategies.

## Figures and Tables

**Figure 1 cells-09-01215-f001:**
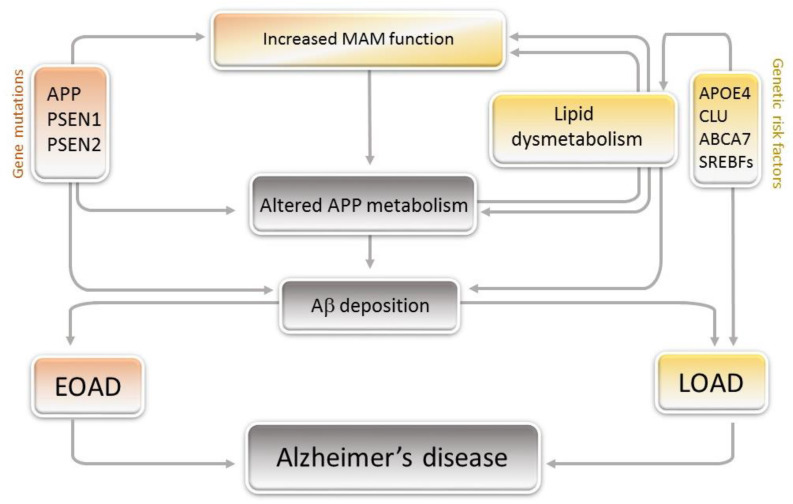
Gene mutations and genetic risk factors linked to lipid dysmetabolism and the progression of Alzheimer’s disease (AD). Gene mutations responsible for inherited early-onset AD cases (EOAD, gene mutations) and genetic risk factors for late-onset AD cases (LOAD, genetic risk factors) lead to altered amyloid precursor protein (APP) processing and brain amyloid-β (Aβ) deposition. Disruption of lipid homeostasis induces abnormal lipid composition in rafts and increased mitochondria-associated endoplasmic reticulum membrane (MAM) function in which targeted APP is proteolytically processed into Aβ by presenilins (PSEN). Conversely, cleavage of APP directly affects cellular lipid composition by altering the synthesis of several lipids that are enriched in lipid rafts. Abbreviations: APOE4 (Apolipoprotein E4); CLU (Clusterin); ABCA7 (ATP-binding cassette sub-family A member 7); SREBFs (Sterol regulatory element-binding genes).

**Figure 2 cells-09-01215-f002:**
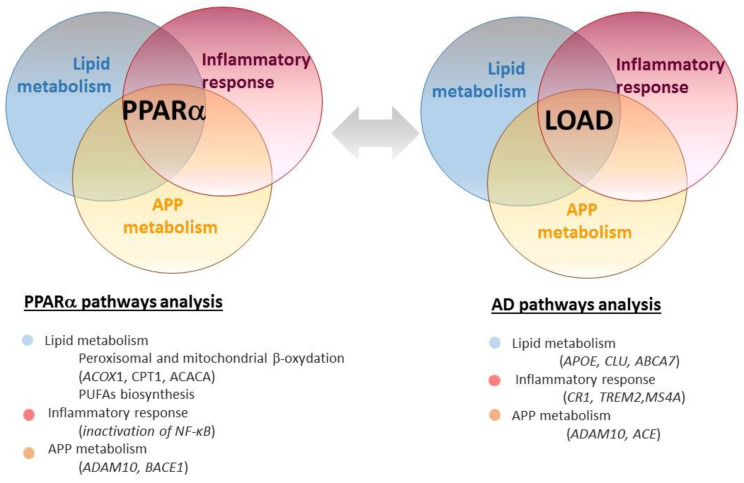
Common pathways regulated by proliferator-activated receptor α (PPARα) and involved in the etiology of AD. Peroxisome proliferator-activated receptor α (PPARα) is a transcription factor that governs pathways involved in the metabolism of lipids, inflammatory response and the metabolism of the amyloid precursor protein (APP), which have been implicated in the etiology of late-onset Alzheimer’s disease (LOAD). Genome-wide association studies identified several genetic risk factors for LOAD, which are involved in pathways that are governed by PPARα. Abbreviations: ABCA7 (ATP-binding cassette subfamily A member 7), ACE (Angiotensin-converting enzyme gene), APOE (Apolipoprotein E), APP (Amyloid precursor protein), ACACA (Acetyl-CoA carboxylase), ACOX1 (Acyl-CoA oxidase), ADAM10 (ADAM Metallopeptidase Domain 10), BACE1 (β-Site amyloid precursor protein-cleaving enzyme 1), CLU (Clusterin), CPT1 (Carnitine palmitoyl transferase), CR1 (Complement receptor 1), NF-κB (Nuclear factor κB), MS4A (Membrane-spanning 4A), PUFAs (Polyunsaturated fatty acids), TREM2 (Triggering receptor expressed on myeloid cells 2).

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
