# Peer review of "Alzheimer’s Disease, a Lipid Story: Involvement of Peroxisome Proliferator-Activated Receptor α"

_cells, 2020, doi:10.3390/cells9051215_

Round 1
Reviewer 1 Report
In this review, Sáez-Orellana et al., describe the evidence which support the notion that impairments in lipid metabolism contribute to the pathology of Alzheimer’s disease. They focus on the roles of nuclear receptors and specifically of the peroxisome proliferator-activated receptor a (PPAR α), in the etiology. While this topic is interesting and the review is timely, several aspects must be improved before acceptance.
Major comments
- In page 2 the authors refer to the “amyloid hypothesis” and state that: “This hypothesis was strengthened by the discovery of mutations in presenilins, the catalytic subunits of the γ-secretase complex leading to an increase in Aβ production [10]”. One of the key recent developments in the Alzheimer’s field is the understanding that different familial cases exhibit dissimilar features, which are very different from these of sporadic Alzheimer’s disease (sAD). A prominent example is the observation that different familial Alzheimer’s disease (fAD)-causing mutations in the sequence of PSEN1 (coding for Presenilin 1), a major component of the γ secretase complex, have opposing effects on the activity of this proteolytic complex. In fact, the majority of fAD-causing mutations lead to reduced activity of the complex γ secretase (Sun et al., Proc Natl Acad Sci U S A.2017 Jan 24;114(4):E476-E485). Furthermore, a few studies, which were focused on the mechanisms that underlie the development of fAD indicated that certain mutations render Presenilin 1 inactive (Xia et al., Neuron. 2015 Mar 4;85(5):967-81 and Ben-Gedalya et al., EMBO J. 2015 Nov 12;34(22):2820-39). Importantly, comparing that Aβ production level in brains of individuals who suffered from sAD to these in brain samples of unaffected controls show similar Aβ productions levels (Szaruga et al., J Exp Med. 2015 Nov 16;212(12):2003-13.). In contrast, the authors found that brain samples of patients who harbored different fAD-causing mutations in PSEN1, exhibited very different levels of Aβ production. In some cases the mutation hyper-activated PSEN1 while in other cases in reduced its activity.
These evidences, which seriously challenge the amyloid hypothesis, should be described and cited to provide the reader with an updated and balanced overview.
- These aforementioned studies also raise the very important question of whether AD is a single disease or a collection of dementia-causing maladies. This question was partially answered by the definition of a new type of neurodegenerative disorder (LATE) that causes dementia but exhibits different features than AD (Nelson et al., Brain. 2019 Jun 1;142(6):1503-1527). LATE patients were previously diagnosed as AD patients. The idea that AD is a collection of maladies that underlie the development of dementia, should be explained.
- An additional crucial aspect that should be comprehensively explained is the relative toxicity rates of different Aβ assemblies. It was shown more than 10 years ago that small Aβ oligomers are the most toxic species. The Selkoe lab reported that dimers are most toxic (Shanker et al., Nat Med. 2008 Aug;14(8):837-42.). Moreover, in some cases protection from Aβ-mediated toxicity is associated with hyper-aggregation (Cohen et al., Cell. 2009 Dec 11;139(6):1157-69.), suggesting that the production of large fibrils could be protective. This aspect should be explained to properly inform the non-expert reader.
- In page 3 an explanation of the links between lipid metabolism and AD appears. They should also describe the roles of lipid contact sites in the development of AD. For example, the mitochondria-associated ER membranes have been linked to the pathology of AD (Pera et al., EMBO J. 2017 Nov 15;36(22):3356-3371). They may also expand the discussion of lipid rafts, such as caveolae, in the development of neurodegeneration. I also think that a figure that illustrated the links of lipids and AD could greatly help the reader.
- At the top of page 9 the authors suggest that female sex hormones protect from AD and state that estrogen replacement therapy is associated with reduced incidence of AD. In the following paragraph they correctly mention that women are at higher risk to develop dementia compared to men. How can they explain this? The statement that differences in life expectancies can partially explain this difference falls short of justifying this apparent discrepancy. Can they provide a speculative explanation of why despite high levels of estrogen women are more susceptible to AD?
Author Response
"Please see the attachment."

Reviewer 2 Report
The manuscript by Sáez-Orellana et al. focusses on the role of PPARs, and markedly PPARα, as a master regulator of energy homeostasis, in Alzheimer's disease.
For this review, the Authors made an appreciable effort to collect data from different groups on several aspects involving PPARα and its natural and synthetic ligands in the pathogenesis and treatment of disease. After a brief introduction to the pathology, the relationship among lipids, physiological cognition and dementia is addressed. Emphasis is given to the involvement of polyunsaturated fatty acids in normal development, plasticity and functioning of the CNS. Then, in a consequential manner, the nuclear receptor superfamily is described, in relation to their involvement in AD, before focussing on PPARs. The Authors provide here a detailed description and original comments on PPAR role in AD-related conditions (obesity, diabetes etc.), emphasising the potential and actual therapies targeting PPARs. Specifically, the Authors focus on PPARα, as the only isoform involved in memory function, and present all major work done with PPARα and its ligands, concerning amelioration of dementia.
Even though a thorough work of critical revision of old and current literature has been accomplished, the Authors fail to extensively discuss the link among PPARs isoforms, which reportedly influence one another in their expression and function. Such interplay, referred by some Authors as the "triad", is also relevant to AD and should be mentioned in the present manuscript.
An additional, minor critique concerns the only figure included in the paper, which appears rather simplistic, and not adequately informative. A more detailed scheme should be provided.
Author Response
"Please see the attachment."

Round 2
Reviewer 1 Report
The authors have addressed my concerns in full. I think that the manuscript can be accepted in its current form.